# Entropy scaling for diffusion coefficients in fluid mixtures

Sebastian Schmitt ⓘ, Hans Hasse & Simon Stephan ⓘ ✉

Entropy scaling is a powerful technique that has been used for predicting transport properties of pure components over a wide range of states. However, modeling mixture diffusion coefficients by entropy scaling is an unresolved task. We tackle this issue and present an entropy scaling framework for predicting mixture self-diffusion coefficients as well as mutual diffusion coefficients in a thermodynamically consistent way. The predictions of the mixture diffusion coefficients are made based on information on the self-diffusion coefficients of the pure components and the infinite-dilution diffusion coefficients. This is accomplished using information on the entropy of the mixture, which is taken here from molecular-based equations of state. Examples for the application of the entropy scaling framework for the prediction of diffusion coefficients in mixtures illustrate its performance. It enables predictions over a wide range of temperatures and pressures including gaseous, liquid, supercritical, and metastable states—also for strongly non-ideal mixtures.

Diffusion in mixtures is important in many natural and technical processes. Numerical values for diffusion coefficients are needed in many applications, e.g., for the design of fluid separation processes, reactors, and combustion processes. However, experimental data on diffusion coefficients are notoriously scarce, so that reliable models for their prediction are needed. We introduce a model type that is based on entropy scaling and enables predictions that were previously infeasible.

In diffusion, two phenomena are distinguished: self-diffusion (a.k.a. tracer diffusion) and mutual (transport) diffusion. Self-diffusion and the corresponding self-diffusion coefficient $D_i$ of a component $i$ describes the Brownian movement of individual particles and is defined for pure components and mixtures. In contrast, mutual diffusion is only defined in mixtures and describes the motion of particle collectives of the different components resulting in macroscopic mass transfer. Obviously, self-diffusion and mutual diffusion are closely related, but there exists no generally applicable relation that connects self-diffusion coefficients and mutual diffusion coefficients. The different diffusion coefficients are schematically depicted in Fig. 1 for a binary mixture.

For describing mutual diffusion[1], there are two common frameworks: those of Fick and Maxwell-Stefan. The corresponding diffusion

coefficients are the Fickian diffusion coefficients $D_{ij}$ (related to the concentration gradient as driving force) and the Maxwell-Stefan diffusion coefficients $Đ_{ij}$ (related to the chemical potential gradient as driving force). The two frameworks are thermodynamically consistent representations and can be transformed into each other. For a binary mixture, these diffusion coefficients are related by

$$D_{ij} = Đ_{ij}\Gamma_{ij}, \tag{1}$$

where $\Gamma_{ij}$ is the thermodynamic factor, which is defined by

$$\Gamma_{ij} = \frac{x_i x_j}{RT}\left(\frac{\partial^2 G}{\partial x_i^2}\right)_{T,p,n_{j\neq i}}, \tag{2}$$

where $x_i$ and $x_j$ are the mole fractions of components $i$ and $j$, respectively, R is the universal gas constant, and $G$ is the Gibbs energy of the mixture.

The two coefficients $D_{ij}$ and $Đ_{ij}$ become equal if the thermodynamic factor is unity, which is the case in the infinite dilution limit (and for ideal mixtures). Furthermore, the self-diffusion coefficient and the mutual diffusion coefficients are related in the infinite dilution limit

Laboratory of Engineering Thermodynamics (LTD), RPTU Kaiserslautern, Kaiserslautern, Germany. ✉e-mail: simon.stephan@rptu.de

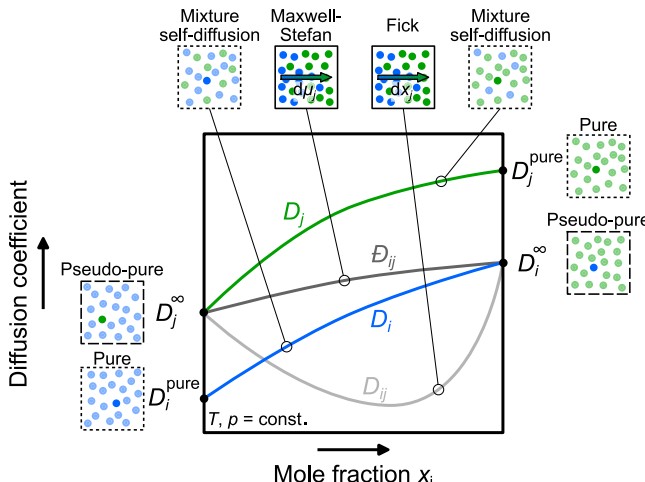

**Fig. 1 | Schematic representation of the different diffusion coefficients in a binary mixture as a function of the mole fraction.** Lines indicate the self-diffusion coefficient of component $i$ $D_i$ (blue) and component $j$ $D_j$ (green), the Maxwell-Stefan diffusion coefficient $Đ_{ij}$ (dark gray), and the Fickian diffusion coefficient $D_{ij}$ (light gray). The limiting cases $D_i^{pure}$, $D_j^{pure}$ (both pure cases) and $D_i^\infty$, $D_j^\infty$ (both pseudo-pure cases) are also depicted.

(cf. Fig. 1). Thus, the following relations apply to the infinite dilution limit:

$$x_i \rightarrow 0: \quad Đ_{ij} = D_{ij} = D_i = D_i^\infty \quad (3)$$

$$x_j \rightarrow 0: \quad Đ_{ij} = D_{ij} = D_j = D_j^\infty \quad (4)$$

where $D_i^\infty$ is the diffusion coefficient of component $i$ infinitely diluted in component $j$, $D_j^\infty$ the diffusion coefficient of component $j$ infinitely diluted in component $i$. Modeling the different diffusion coefficients in a mixture in a consistent way is a challenging task. In this work, we propose a methodology that provides such a framework.

Physical models for predicting mixture diffusion coefficients at gaseous states are known and established for a long time within kinetic gas theory[2]. The prediction of mixture diffusion coefficients at states where significant intermolecular interactions are present, on the other hand, is still an unresolved problem. Einstein has proposed a method for estimating infinite-dilution diffusion coefficients in liquids[3], for which several modifications exist today[2,4,5]. For estimating the concentration dependence of mutual diffusion coefficients in mixtures, several empirical models have been proposed, e.g., the Vignes model and the generalized Darken model[6–8]. However, these models often fail for strongly non-ideal mixtures (see Suppl. Note 8).

In recent years, entropy scaling has received significant attention for modeling transport properties[9–11]. It is based on the discovery of Rosenfeld[12,13], that dynamic properties (i.e., viscosity, thermal conductivity, and self-diffusion coefficient) of pure components, when properly scaled by the density and temperature, are a monovariate function of the configurational entropy (sometimes also referred to as "residual entropy" or "excess entropy"). This scaling behavior is physically based and related to isomorph theory[11,14]. After being rediscovered by Dyre in a seminal review[11], entropy scaling has become a popular approach in the last ten years. For modeling transport properties, the entropy scaling principle has been cast into many executable models using several different modified scaling approaches[9,15–19]. For obtaining the entropy at a desired state point (e.g., given by $T, p$), usually an equation of state (EOS) is used. Entropy scaling can be favorably combined with molecular-based EOS[20], which enables predictions beyond the available data[17,21]. Entropy scaling is well established for predicting the viscosity and thermal conductivity of mixtures, for example, demonstrated by Gross and co-workers[9,22] and Bell and co-workers[23,24] in recent years. The corresponding models are based on combination and mixing rules and often enable a reliable prediction of mixture viscosities and thermal conductivities[17,23,25]. However, entropy scaling for mixture diffusion coefficients—as depicted in Fig. 1—has not yet been developed. So far, only the pure component limiting cases for the self-diffusion coefficients $D_1^{pure}$ and $D_2^{pure}$ (cf. Fig. 1) can be described by entropy scaling models from the literature[16–19,26]. There are few molecular simulation studies of self-diffusion coefficients in mixtures available in the literature[10,27–33] in which also the entropy scaling behavior of the data was considered. They evaluate either simple model systems or special cases like metallic fluids. They show that self-diffusion coefficients of such fluids can follow a universal monovariate behavior, but the authors do not provide models for describing the self-diffusion coefficients. Specifically, a quasi-universal scaling law for mixture diffusion coefficients has been proposed by Bell and Dyre[10]—however, being limited to self-diffusion. Truskett and co-workers[29–32] have studied the monovariate scaling behavior based on computer experiment data. Similarly, Fertig et al.[33] have demonstrated that elements of monovariate scaling are also present in model mixture diffusion coefficient data. However, in none of these studies a generally applicable modeling framework for consistently describing the different diffusion coefficients in mixtures has been developed, so that this is still an unresolved issue.

In this work, we propose an entropy scaling model for predicting mixture diffusion coefficients, namely the self-diffusion coefficients as well as the (Fickian and Maxwell-Stefan) mutual diffusion coefficients, without any adjustable mixture parameters. The physical framework we have developed for predicting diffusion coefficients in mixtures can be characterized as follows: (1) It can be applied in the entire fluid region, i.e. it covers gases, liquids, and supercritical fluids, phase equilibria, and even metastable states. (2) It describes both self-diffusion and mutual diffusion in a consistent way. (3) It comprises the dependence of the diffusion coefficients on temperature, pressure, and composition in the entire fluid region. The applicability of the model is demonstrated for binary mixtures in this work.

The approach developed in this work is based on three central ideas and concepts: (1) Infinite-dilution diffusion coefficients are treated as pseudo-pure components that exhibit a monovariate scaling behavior, which can be treated by classical entropy scaling[17]. (2) Therefore, the mixture diffusion coefficient limiting cases, i.e., the pure component and pseudo-pure component self-diffusion coefficients (cf. Fig. 1), are modeled as functions of the entropy. This requires at least one reference data point for each limiting case. (3) Based on the information of the limiting cases, the concentration dependence of the diffusion coefficients $D_i$, $D_j$, $Đ_{ij}$, and $D_{ij}$ are predicted using combination and mixing rules. (4) It correctly captures the limits of the different diffusion coefficients to the pure component and infinite dilution. We demonstrate that these predictions can succeed without any adjustable mixture parameter. The performance of the approach is demonstrated using model fluids as well as real substance systems. In this work, we employ molecular-based EOS models. Yet, also other EOS types such as multiparameter[34] or cubic EOS[35] could be used.

## Results

First, we demonstrate the applicability of a monovariate scaling to the infinite dilution self-diffusion coefficients by treating $D_i^\infty$ as a pseudo-pure component property. This enables the prediction of $D_i^\infty$ to practically all fluid states based on very little data. The monovariate scaling of the infinite-dilution diffusion coefficients uncovered in this work provides the basis for the modeling of the different mixture diffusion coefficients shown in the second part of this section. The applicability of the modeling approach is demonstrated in the second part of this section, where model predictions for the self-diffusion as

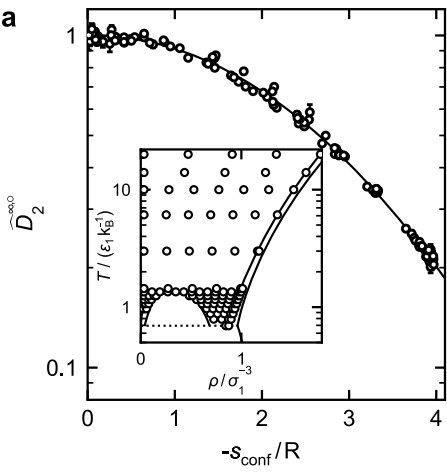

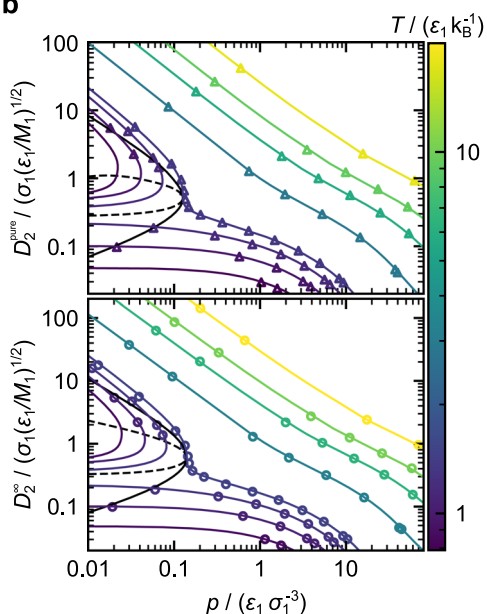

**Fig. 2 | Entropy scaling of diffusion coefficients in a Lennard-Jones mixture with** $\sigma_2 = \sigma_1$, $\varepsilon_2 = 0.9\,\varepsilon_1$, **and** $\varepsilon_{12} = 1.2\,\sqrt{\varepsilon_1\varepsilon_2}$. **a** Scaled infinite-dilution diffusion coefficient of component 2 $\widehat{D}_2^{\infty,\circ}$ as a function of the reduced configurational entropy $s_{\text{conf}}/R$. The line indicates the entropy scaling model. Symbols are MD simulation data from this work. The inset shows the simulation state points in the temperature-density phase diagram of component 1. Therein, solid lines indicate the phase envelopes from refs. 46,47. **b** Self-diffusion coefficient of component 2 $D_2^{\text{pure}}$ (top) and infinite-dilution diffusion coefficient of component 2 $D_2^{\infty}$ (bottom) as a function of the pressure. Lines are the entropy scaling model and symbols are simulation results from ref. 17 ($D_2^{\text{pure}}$) and from this work ($D_2^{\infty}$). The black solid line indicates the vapor–liquid equilibrium, and the black dashed line is the corresponding spinodal. Colors indicate the temperature. Source data are provided as a Source Data file.

well as mutual diffusion coefficients are compared to reference data for model and real substance systems.

## Infinite-dilution diffusion coefficients

Figure 2a demonstrates the monovariate scaling behavior for infinite-dilution diffusion coefficients $\widehat{D}_2^{\infty,\circ}$ for a binary Lennard-Jones mixture.

Results are shown for the scaled self-diffusion coefficient of component 2 at infinite dilution $\widehat{D}_2^{\infty,\circ}$; in our notation, the hat ^ and ˚ refer to specific parts of the scaling, see Methods. The data collapse to a monovariate curve, which is impressive considering the fact that a large range of states was studied, cf. Fig. 2a-inset. The quality of the scaling of $\widehat{D}_2^{\infty,\circ}$ (i.e., how well the data can be described by a

monovariate function) is essentially the same as that found for the pure component diffusion coefficient $D_2$ (see Suppl. Note 5 for quantification). This supports the picture introduced in this work that $D_2^{\infty}$ can be considered as a pseudo-pure component property. This picture is physically meaningful considering the fact that in both cases, the mobility of a single particle in a homogeneous environment is described. The Chapman-Enskog diffusion coefficient agrees well with the low-density simulation results—as indicated by the convergence of $\widehat{D}_2^{\infty,\circ} \to 1$ for $s_{\text{conf}}/R \to 0$.

Figure 3 shows the simulation results for the infinite-dilution diffusion coefficients in the systems acetone + isobutane and ethanol + chlorine.

The simulations comprise states in the gas, liquid, and supercritical regions. Details on the simulations are provided in Suppl. Note 2. For both systems, the infinite-dilution diffusion coefficient shows a monovariate behavior with respect to the configurational entropy confirming the results for the Lennard-Jones model system. Significant deviations from the monovariate behavior are observed for the real systems only for $\tilde{s}_{\text{conf}} < 1$, i.e. for low-density state points. This might be due to larger uncertainties of the simulations in this region.

In Suppl. Note 6, the results for two more binary Lennard-Jones systems and a third real substance system are presented that support the findings discussed here. For the Lennard-Jones model systems, the "global" parameters from ref. 17 were sufficient for describing $\widehat{D}_2^{\infty,\circ}$. This finding was not expected and demonstrates the broad applicability of the universal parameters $g_1$ and $g_2$ in the correlation that were determined in ref. 17.

From the models for the pure component self-diffusion coefficient $\widehat{D}_2^{\text{pure},\circ}(\tilde{s}_{\text{conf}})^{17}$ and the infinite-dilution diffusion coefficient $\widehat{D}_2^{\infty,\circ}(\tilde{s}_{\text{conf}})$, both $D_2^{\text{pure}}$ and $D_2^{\infty}$ can be predicted in a wide range of states. For the Lennard-Jones systems, the configurational entropy was taken from the Kolafa-Nezbeda EOS[36], which is known to give an accurate the description of Lennard-Jones fluids[37,38]. We used the mixture implementation for the Kolafa-Nezbeda EOS from refs. 17,37. Figure 2b shows the results of the entropy scaling models for $D_2$ and $D_2^{\infty}$ in comparison to the molecular dynamics (MD) simulation results. Both methods are in good agreement, which is a result of the good performance of the EOS as well as the fact that both properties show a monovariate relation with respect to the configurational entropy.

Hence, reliable analytical models for the limiting cases $D_1$, $D_2$, $D_1^{\infty}$, and $D_2^{\infty}$ (cf. Fig. 1) are now available that can be applied in all fluid regions. They are the basis for the next step, the prediction of $D_1$, $D_2$, $\text{Đ}_{12}$, and $D_{12}$ at arbitrary compositions in the mixture.

## Diffusion coefficients in mixtures

The model proposed in this work (see Methods) is able to predict all diffusion coefficients in binary mixtures. Figure 4 shows the model predictions for all four diffusion coefficients ($D_1$, $D_2$, $\text{Đ}_{12}$, $D_{12}$) of three Lennard-Jones systems for the entire composition range and different temperatures. For comparison, simulation data from ref. 39 are used. The predictions from the entropy scaling framework and the computer experiment results agree well for all investigated systems and all four diffusion coefficients (see Fig. 4).

For both self-diffusion coefficients, for which the simulation results have a smaller statistical uncertainty than for the mutual diffusion coefficients, the entropy scaling model describes most simulation data within their uncertainty, despite the strong non-ideality that leads to extrema at about $x_2 \approx 0.4$ mol mol⁻¹ (a–d in Fig. 4) or even to the occurrence of a liquid-liquid equilibrium gap (i–l in Fig. 4)—the phase behavior of these model systems was comprehensively studied in ref. 40. The simulation data for $\text{Đ}_{12}$ scatter more and have larger error bars than those for $D_1$ and $D_2$. The entropy scaling framework predicts the $\text{Đ}_{12}$ simulation data very well in the entire composition range. In particular, the strongly non-ideal behavior is captured accurately by the entropy scaling framework. We have compared the

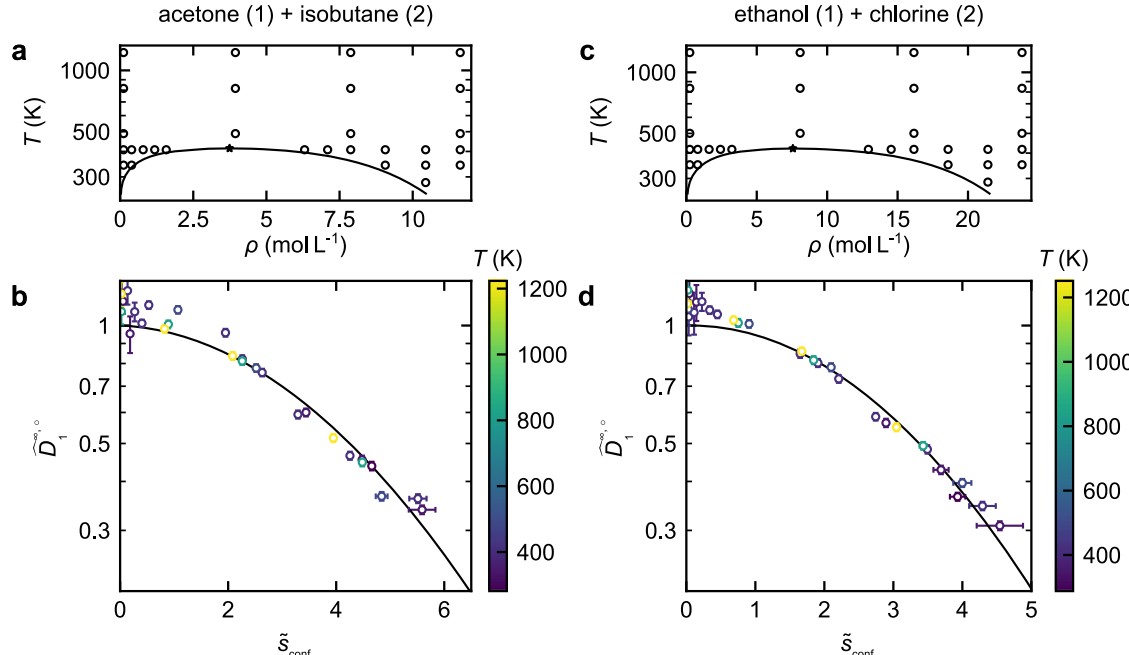

**Fig. 3 | Scaling behavior of the infinite-dilution diffusion coefficients in the system acetone (1) + isobutane (2) (left) and ethanol (1) + chlorine (2).** **a, c** Studied state points (symbols) in the temperature-density phase diagram of the solvents. The line indicates the vapor–liquid equilibrium and the star the critical point as calculated from the PC-SAFT EOS[41]. **b, d** Scaled infinite-dilution diffusion coefficient of the solutes $\widehat{D}_1^{\infty,\circ}$ as a function of the reduced configurational entropy $\tilde{s}_{\mathrm{conf}}$. The symbols are the simulation results (color indicates the temperature) and the line the entropy scaling model (fitted to the simulation results). The error bars represent the uncertainties of the molecular simulation results. Source data are provided as a Source Data file.

entropy scaling framework presented in this work to results from the established Vignes and Darken model (see Suppl. Note 8). The entropy scaling model outperforms the empirical models. Also the temperature and pressure dependency of the diffusion coefficients are very well described by the model (cf. Fig. 4 and Suppl. Note 7, respectively). For the predictions of the Fickian diffusion coefficient, the trends are correctly predicted by the entropy scaling predictions, but deviations are observed, cf. Fig. 4d,h,l. The deviations between simulation results and entropy scaling are larger for the Fickian diffusion coefficient, which can be attributed to the inclusion of the thermodynamic factor adding additional complexity to the model predictions (and the MD sampling[33]). In contrast to the other three diffusion coefficients ($D_1$, $D_2$, $\Bar{D}_{12}$), the Fickian diffusion coefficient $D_{12}$ exhibits the opposite extremum behavior, which is a result of the underlying thermodynamic factor behavior. Due to the coupling of entropy scaling and EOS, the model is inherently consistent and can be applied over a wide range of states. As an example, Fig. 4i, j includes a system with weak cross-interactions resulting in a liquid-liquid equilibrium at the considered conditions. The entropy scaling model proposed in this work is able to predict not only the different diffusion coefficients in the bulk phases, but also describes the diffusion coefficients of the coexisting phases of the liquid-liquid equilibrium. Also, at the upper critical solution point, the thermodynamic factor becomes zero and so does the Fickian diffusion coefficient ($x_2 \approx 0.38\,\mathrm{mol\,mol}^{-1}$). This is as expected for the Fickian diffusion coefficient at a critical point[37] and correctly captured by the model.

Figure 5 shows the results for the self-diffusion coefficients in three real substance systems: $n$-hexane + $n$-dodecane, acetone + chloroform, and nitrobenzene + $n$-hexane. Figure 6 shows entropy scaling predictions for the Fickian diffusion coefficient in three real substance systems: toluene + $n$-hexane, acetone + chloroform, and toluene + acetonitrile. For all systems, only very limited experimental data are available (this is common for practically all real mixtures), which is used for the validation of the predictions. The thermodynamic properties of the systems were modeled by the PC-SAFT EOS[41], and the

component-specific models for the pure substances were taken from the literature (see Suppl. Note 3 for details). The Berthelot combination rule parameter $\xi_{12}$ for the systems acetone + chloroform and nitrobenzene + $n$-hexane were adjusted to match the respective vapor–liquid equilibria qualitatively.

The entropy scaling predictions of both self-diffusion coefficients and the experimental data are in good agreement. Only for the system acetone + chloroform, some deviations are observed. For the systems $n$-hexane + $n$-dodecane and nitrobenzene + $n$-hexane, the model predicts the experimental data very well; the mean relative deviations are $\overline{\delta D_1} = 1.24\,\%$ and $\overline{\delta D_2} = 2.91\,\%$ ($n$-hexane + $n$-dodecane) and $\overline{\delta D_1} = 4.96\,\%$ (nitrobenzene + $n$-hexane), respectively. For the system $n$-hexane + $n$-dodecane, the two components strongly differ in their molar mass, which yields a non-linear behavior of $D_1(x_2)$ and $D_2(x_2)$. This behavior is predicted well by the framework. Also, for the system nitrobenzene + $n$-hexane, the non-ideality of the diffusion coefficients is well captured by the model. For the system acetone + chloroform, the mean relative deviations are $\overline{\delta D_1} = 8.58\,\%$ and $\overline{\delta D_2} = 9.69\,\%$.

Figure 6 shows the results for the Fickian diffusion coefficient $D_{12}$ in three systems, namely toluene + $n$-hexane, acetone + chloroform, and toluene + acetonitrile. All results are at ambient pressure, but at different temperatures and shown as a function of the mole fraction. The Fickian diffusion coefficient increases with increasing temperature. The entropy scaling predictions are in good agreement with the experimental data ($\overline{\delta D_{12}} = 3.02\,\%$ for toluene + $n$-hexane, $\overline{\delta D_{12}} = 3.67\,\%$ for acetone + chloroform, and $\overline{\delta D_{12}} = 10.27\,\%$ for toluene + acetonitrile). For the system toluene + $n$-hexane, the entropy scaling model yields good agreement. For the systems toluene + acetonitrile and acetone + chloroform, some deviations are observed–especially in the vicinity of the extremum values.

Figure 7 shows the predictions for the Maxwell-Stefan diffusion coefficient phase diagrams of the system toluene + $n$-hexane at two pressures: $p = 0.1\,\mathrm{MPa}$ (a) and $p = 4\,\mathrm{MPa}$ (b). The diffusion coefficients of the coexisting phases can be obtained in a straightforward manner by the framework. Additional results (including supercritical,

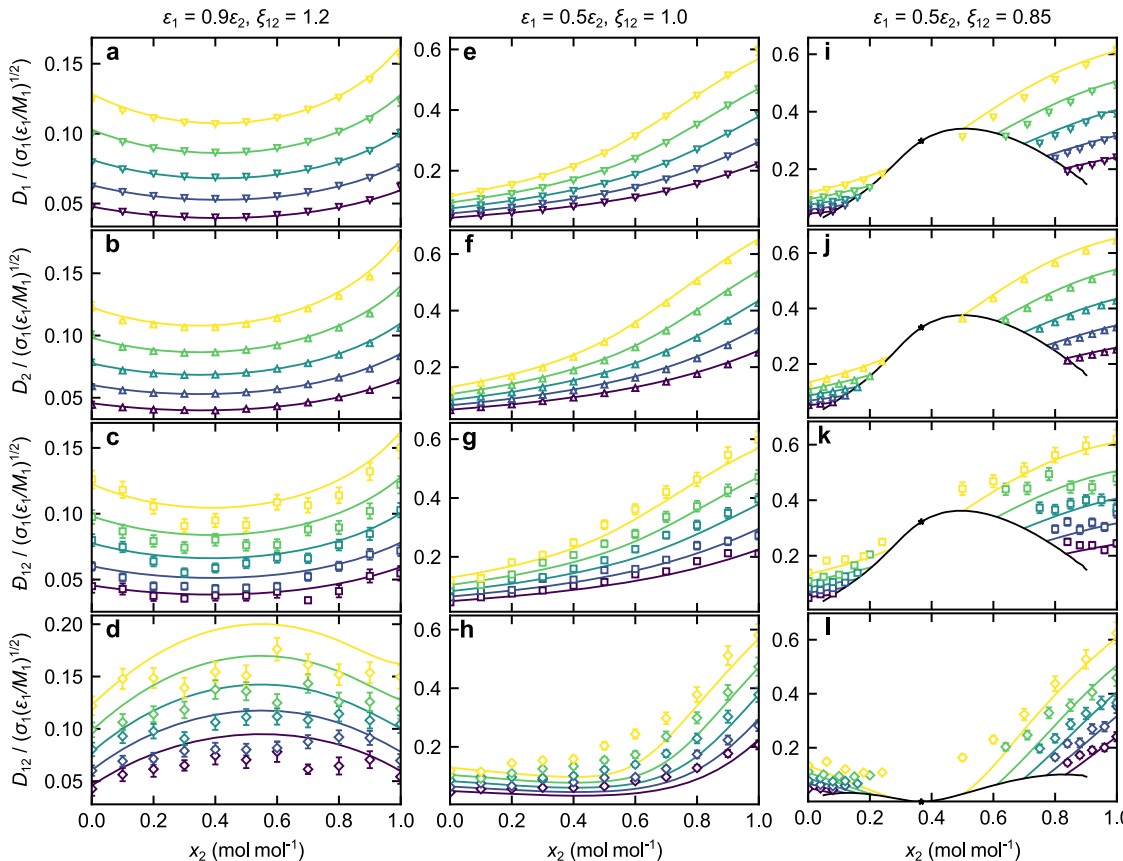

**Fig. 4 | Diffusion coefficients in Lennard-Jones mixtures.** Diffusion coefficients in three binary Lennard-Jones mixtures with $\sigma_2 = \sigma_1$ as a function of the mole fraction $x_2$ at $p = 0.13\,\sigma_1^3\varepsilon_1^{-1}$ (**a–d**) and $p = 0.26\,\sigma_1^3\varepsilon_1^{-1}$ (**e–l**). Energy parameters $\varepsilon_1$, $\varepsilon_2$, and mixing parameter $\xi_{12}$ are given for each column. **a, e, i** Self-diffusion coefficient of component 1 $D_1$; **b, f, j** Self-diffusion coefficient of component 2 $D_2$; **c, g, k** Maxwell-Stefan diffusion coefficient $\Delta_{12}$; **d, h, l** Fickian diffusion coefficient $D_{12}$. Lines are predictions from the entropy scaling framework. Symbols are simulation results from ref. [39]. The entropy scaling framework was used in combination with the Kolafa-Nezbeda EOS[36]. The colors indicate the temperature $T \in \{0.79, 0.855, 0.92, 0.985, 1.05\}\,k_B\varepsilon_1^{-1}$ (from blue to yellow). The black lines indicate the liquid-liquid equilibrium, and the star is the critical point (only **i–l**). The error bars represent the simulation uncertainty given in ref. [39]. Source data are provided as a Source Data file.

metastable, and unstable states) are presented in Suppl. Note 10. At $p = 0.1$ MPa, both components are subcritical (see inset in Fig. 7a). The Maxwell-Stefan diffusion coefficient is shown for three isotherms (340, 360, 385 K). As shown in the inset (see Fig. 7a), one isotherm is entirely in the liquid phase (340 K), one isotherm passes through the vapor–liquid equilibrium (360 K), and one isotherm is entirely in the gas phase (385 K). The Maxwell-Stefan diffusion coefficient in the gas phase is significantly larger than that in the liquid phase. In the liquid phase, the diffusion coefficient strongly depends on the composition. In the gas phase, the Maxwell-Stefan diffusion coefficient exhibits only a minor dependency on the composition. This is in line with the Chapman-Enskog theory[2]—which is inherently incorporated into our framework. At $p = 4$ MPa (cf. Fig. 7b), $n$-hexane is supercritical, and a critical point exists at about $T = 547.18$ K (see inset). Four isotherms are plotted: One subcritical isotherm ($T = 540$ K), the critical isotherm ($T = 547.18$ K), one isotherm passing through the vapor–liquid equilibrium ($T = 570$ K), and one isotherm entirely in the gas/supercritical phase ($T = 590$ K). The isotherm at $T = 540$ K exhibits a transition between the two pseudo-pure component limiting cases—from a liquid state at $x_2 = 0$ to a supercritical state at $x_2 = 1$ mol mol$^{-1}$. At the critical point, the diffusion coefficient exhibits a large gradient with respect to the composition. Due to the thermodynamic consistency established by the EOS, the framework correctly predicts the Fickian diffusion coefficient to be zero at critical points (see Fig. 4).

In Suppl. Note 10, it is shown that also diffusion coefficients of metastable and unstable states as well as at spinodal states can be

described by the framework (which is for example relevant for nucleation). This significantly exceeds the capabilities of all presently available diffusion coefficient models.

## Discussion

In this work, a methodology for predicting diffusion coefficients in mixtures in a consistent manner for all fluid states is proposed. This includes self-diffusion and mutual diffusion coefficients. The approach combines several physical concepts, including the modified Rosenfeld scaling, the Chapman-Enskog theory, and a molecular-based EOS. The framework enables predictions for the mixture diffusion coefficients without any adjustable parameters based on the limiting cases of the pure components and pseudo-pure components (i.e. infinite dilution). Due to the coupling of entropy scaling and an EOS, the framework can consistently describe diffusion coefficients in mixtures in different phases (gas, liquid, supercritical, metastable)—including coexisting phases such as vapor–liquid and liquid-liquid equilibria. Even predictions of regions in which no data are available are possible. The strong predictive capabilities are a result of the physical backbone of the framework. The fact that infinite-dilution diffusion coefficients exhibit a monovariate scaling behavior was uncovered here for the first time. It enables an efficient entropy scaling modeling of the limiting mixture diffusion coefficient cases based on very few data. Since the limiting case diffusion coefficients are monovariate functions with respect to the entropy, extrapolations beyond the

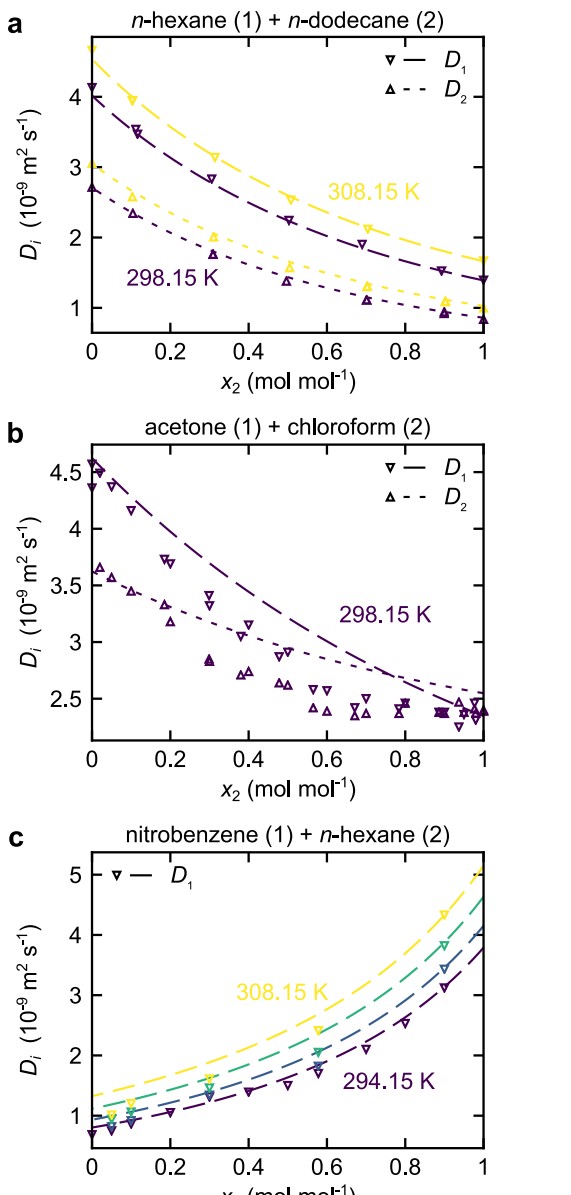

**Fig. 5 | Self-diffusion coefficients $D_i$ of real substance mixtures predicted by the entropy scaling model as a function of the mole fraction $x_2$ at $p$ = 0.1 MPa.** Symbols are experimental data from the literature and lines are model predictions. **a** Mixture $n$-hexane (1) + $n$-dodecane (2); experimental data from ref. 48. **b** Mixture acetone (1) + chloroform (2); experimental data from ref. 49. **c** Mixture nitrobenzene (1) + $n$-hexane (2); experimental data from ref. 50. Source data are provided as a Source Data file.

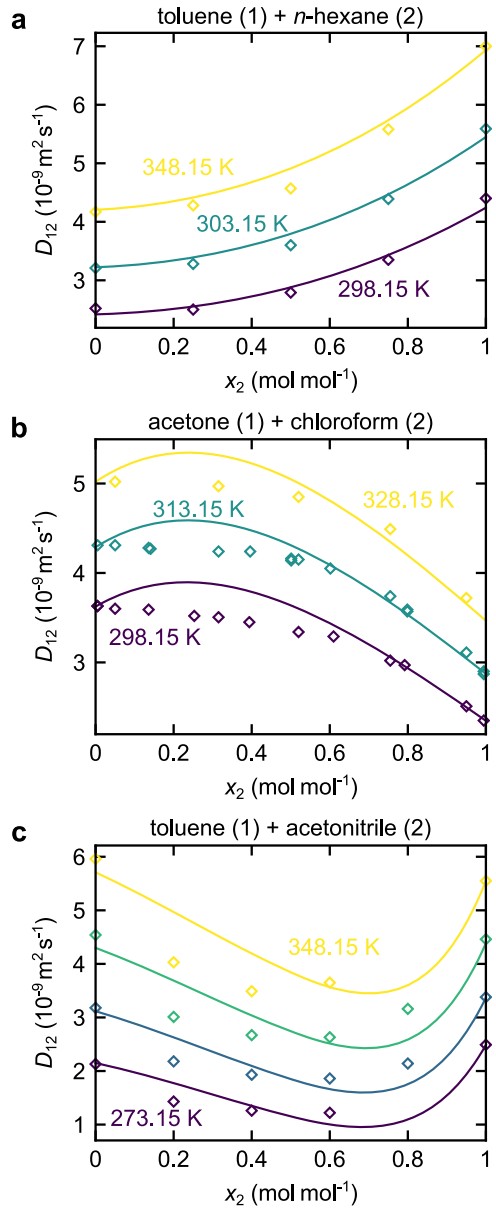

**Fig. 6 | Fickian diffusion coefficients of real substance mixtures predicted by the entropy scaling model as a function of the mole fraction $x_2$ at $p$ = 0.1 MPa.** Symbols are experimental data, and lines are model predictions. **a** Mixture toluene (1) + $n$-hexane (2); experimental data from ref. 51. **b** Mixture acetone (1) + chloroform (2); experimental data from refs. 52,53. **c** Mixture toluene (1) + acetonitrile (2); experimental data from ref. 54. Source data are provided as a Source Data file.

temperature and pressure range where data are available are feasible. In the mixture, no general monovariate behavior retains. Nevertheless, the mixture entropy, in combination with mixing and combination rules, enables predictions of mixture diffusion coefficients, which was demonstrated here for model mixtures as well as real substance systems.

The entropy scaling framework requires an accurate and robust EOS model that describes mixture thermodynamic properties reliably, i.e. the phase equilibria, the entropy, the thermodynamic factor, and the second virial coefficient. If such a model is available, the entropy scaling framework proposed in this work is a powerful tool. For future work, the extension to multi-component systems, e.g., refs. 42–44, would be interesting.

## Methods

The framework proposed in this work consists of two elements: (1) Treating the infinite dilution self-diffusion coefficient as a pseudo-pure component to obtain a monovariate scaling; (2) The application of mixing and combination rules for predicting the different diffusion coefficients in a mixture. Throughout this work, the configurational entropy $s_{\text{conf}}(T, \rho, \underline{x})$ is calculated from molecular-based EOS as the derivative of the configurational Helmholtz energy $a_{\text{conf}}$ with respect to temperature $T$ at constant volume $v$ and composition $\underline{x}$, i.e.

$$s_{\text{conf}} = -\left(\frac{\partial a_{\text{conf}}}{\partial T}\right)_{v, \underline{x}}. \tag{5}$$

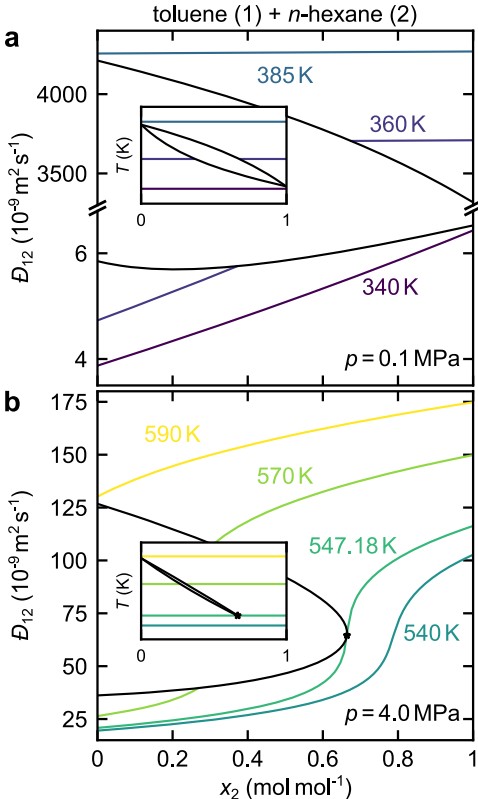

**Fig. 7 | Prediction of the Maxwell-Stefan diffusion coefficients in real mixtures over a wide range of conditions.** Maxwell-Stefan diffusion coefficient $Ð_{12}$ in the mixture toluene (1) + $n$-hexane (2) as a function of the mole fraction of $n$-hexane $x_2$ at $p = 0.1$ MPa (**a**) and $p = 4$ MPa (**b**). Black lines: Vapor-liquid equilibrium; colored lines: Isotherms. Insets show the $T - x$ vapor–liquid-equilibrium phase diagram at the respective pressure. Source data are provided as a Source Data file.

Therein, the configurational Helmholtz energy $a_{conf}$ is defined as $a_{conf} = a - a_{ideal}$ with $a_{ideal}$ being the Helmholtz energy of the ideal gas. We use a reduced configurational entropy defined as

$$\tilde{s}_{conf} = -s_{conf}/(\mathrm{R}m), \tag{6}$$

where $m$ is the segment parameter of molecular-based EOS.

## Entropy scaling of infinite-dilution diffusion coefficients

The infinite-dilution diffusion coefficient $D_i^\infty(T,p)$ is treated as a pseudo-pure component property such that the corresponding scaled property $\widehat{D}_i^{\infty,\circ}(\tilde{s}_{conf})$ exhibits a monovariate function with respect to the reduced configurational entropy $\tilde{s}_{conf}$ (cf. Eq. (6)). The modeling methodology is described in the following. There are many scaling approaches available in the literature that aim at establishing a monovariate behavior of transport coefficients. Here, we use the scaling described in ref. 17. The pure component scaling described in ref. 17 (which follows refs. 13,16) is adapted here to model the pseudo-pure component diffusion coefficients at infinite dilution. Thus, we propose that the infinite-dilution diffusion coefficient is transformed using

$$D_i^{\infty,\circ} = D_i^\infty \frac{\rho_N^{1/3}}{\sqrt{\mathrm{R}T/M_{CE}}} \left(\frac{-s_{conf}}{\mathrm{R}}\right)^{2/3}, \tag{7}$$

where $\rho_N$ is the number density of the solvent at given $T$ and $p$. The reference mass $M_{CE}$ is adopted from the Chapman-Enskog (CE) theory as

$$M_{CE} = \frac{2}{1/M_i + 1/M_j}, \tag{8}$$

where $M_i$ and $M_j$ are the molar masses of the pure components. The modified Rosenfeld-scaled diffusion coefficient $D_i^{\infty,\circ}$ exhibits some scattering in the zero-density limit for $s_{conf} \to 0$. Therefore, the scaling is further modified using the Chapman-Enskog diffusion coefficient $D_{CE,i}^{\infty,\circ}$, which is computed using a Lennard-Jones kernel (see Suppl. Note 1 for details). Finally, the CE-scaled infinite-dilution diffusion coefficient $\widehat{D}_i^{\infty,\circ}$ is described by a split between the low-density (LD) and the high-density (HD) region as

$$\widehat{D}_i^{\infty,\circ} = \underbrace{\frac{D_i^{\infty,\circ}}{D_{CE,i}^{\infty,\circ}} W(\tilde{s}_{conf})}_{\text{LD}} + \underbrace{\frac{D_i^{\infty,\circ}}{\min(D_{CE,i}^{\infty,\circ})}(1 - W(\tilde{s}_{conf}))}_{\text{HD}}, \tag{9}$$

where $W$ is a smoothed transition function given by $W(\tilde{s}_{conf}) = 1/(1 + \exp(20(\tilde{s}_{conf} - \tilde{s}_{conf}^\times)))$ with $\tilde{s}_{conf}^\times = 0.5$. The function $W$ establishes a smooth transition from the low-density region to the high-density region. The choice of $\tilde{s}_{conf}^\times$ is heuristic. The resulting CE-scaled infinite-dilution diffusion coefficient can be described by a simple continuous, monovariate function $\widehat{D}_i^{\infty,\circ} = \widehat{D}_i^{\infty,\circ}(\tilde{s}_{conf})$. Here, a function with two adjustable, system-dependent parameters $\alpha_2^{(D_i^\infty)}$ and $\alpha_3^{(D_i^\infty)}$ was used given as

$$\ln\left(\widehat{D}_i^{\infty,\circ}\right) = \frac{\alpha_2^{(D_i^\infty)}(\tilde{s}_{conf})^2 + \alpha_3^{(D_i^\infty)}(\tilde{s}_{conf})^3}{1 + g_1^{(D)}\ln(\tilde{s}_{conf} + 1) + g_2^{(D)}\tilde{s}_{conf}}, \tag{10}$$

where $g_1^{(D)} = 0.6632$ and $g_2^{(D)} = 9.4714$ are global parameters fitted to the self-diffusion coefficient of the Lennard-Jones fluid[17]. The system-dependent parameters $\alpha_2^{(D_i^\infty)}$ and $\alpha_3^{(D_i^\infty)}$ were fitted to reference data of the pseudo-pure component, i.e. the diffusion coefficients at infinite dilution. This scaling approach was tested using model systems and real substance systems (see Results). Therefore, MD simulations were carried out using the simulation engine ms2[45].

## Predicting diffusion coefficients in mixtures

For predicting the diffusion coefficients $D_i(x_j)$, $D_j(x_j)$, $Ð_{ij}(x_j)$, and $D_{ij}(x_j)$ in a mixture, the entropy scaling framework is extended using combining and mixing rules. Diffusion coefficients in mixtures are predicted based on the limiting case models, i.e. the models for the self-diffusion coefficient of the pure components and the (infinite dilution) pseudo-pure component. The self-diffusion coefficient $D_i$ in a mixture is calculated using the limiting case entropy scaling model of the pure component self-diffusion coefficient $D_i^{pure}$ and the limiting case entropy scaling model of the infinite-dilution diffusion coefficient $D_i^\infty$ in the solvent $j$. The Maxwell-Stefan diffusion coefficient is calculated using both infinite-dilution diffusion coefficient limiting case entropy scaling models $D_i^\infty$ and $D_j^\infty$. Therefore, the scaled mixture diffusion coefficient $\widehat{\Lambda}^\circ \in \{\widehat{D}_i^\circ, \widehat{D}_j^\circ, \widehat{Ð}_{ij}^\circ\}$ is computed as

$$\ln\left(\widehat{\Lambda}^\circ\right) = \frac{\alpha_2^{(\Lambda)}(\tilde{s}_{conf})^2 + \alpha_3^{(\Lambda)}(\tilde{s}_{conf})^3}{1 + g_1^{(D)}\ln(\tilde{s}_{conf} + 1) + g_2^{(D)}\tilde{s}_{conf}}, \tag{11}$$

where $\tilde{s}_{conf}$ is the scaled configurational entropy of the mixture, i.e. $\tilde{s}_{conf} = s_{conf}/(\mathrm{R}m_{mix})$ with $m_{mix} = x_i m_i + x_j m_j$. The parameters $\alpha_k^{(\Lambda)}$ (with $k \in \{2, 3\}$) are calculated using linear mixing rules

$$\alpha_k^{(D_i)} = x_i \alpha_k^{(D_i^{pure})} + x_j \alpha_k^{(D_i^\infty)}, \tag{12}$$

$$\alpha_k^{(D_j)} = x_i \alpha_k^{(D_j^\infty)} + x_j \alpha_k^{(D_j^{\text{pure}})}, \tag{13}$$

$$\alpha_k^{(\text{Đ}_{ij})} = x_i \alpha_k^{(D_j^\infty)} + x_j \alpha_k^{(D_i^\infty)}, \tag{14}$$

for the self-diffusion coefficients of components $i$, the self-diffusion coefficient of component $j$, and the Maxwell-Stefan diffusion coefficient, respectively.

The final (unscaled) diffusion coefficient $\Lambda \in \{D_i, D_j, Đ_{ij}\}$ is calculated as

$$\Lambda = \frac{\hat{\Lambda}^\circ}{\frac{W(\bar{s}_{\text{conf}})}{\Lambda_{\text{CE}}^\circ} + \frac{1 - W(\bar{s}_{\text{conf}})}{\min(\Lambda_{\text{CE}}^\circ)}} \cdot \sqrt{\frac{RT}{M_{\text{ref}}^\Lambda}} \frac{1}{\rho_N^{1/3}} \left( \frac{-s_{\text{conf}}}{R} \right)^{-2/3}, \tag{15}$$

where $\Lambda_{\text{CE}}^\circ$ is the Chapman-Enskog diffusion coefficient of the mixture, $\rho_N$ is the number density of the mixture, and $M_{\text{ref}}^\Lambda$ is the reference mass of the mixture, which is calculated as

$$M_{\text{ref}}^{D_i} = x_i M_i + x_j M_{\text{CE}}, \tag{16}$$

$$M_{\text{ref}}^{D_j} = x_i M_{\text{CE}} + x_j M_j, \quad \text{and} \tag{17}$$

$$M_{\text{ref}}^{Đ_{ij}} = M_{\text{CE}} \tag{18}$$

for the self-diffusion coefficients $D_i$ and $D_j$ and the Maxwell-Stefan diffusion coefficient $Đ_{ij}$, respectively. Details on the calculation of the Chapman-Enskog property of the mixture $\Lambda_{\text{CE}}^\circ$, the reference mass $M_{\text{ref}}$, and $\hat{\Lambda}^\circ = \hat{\Lambda}^\circ(s_{\text{conf}}^{\text{mix}}, \alpha_{k,\text{mix}})$ are given in Suppl. Note 1.

The Fickian diffusion coefficient is computed from the Maxwell-Stefan diffusion coefficient by the thermodynamic factor as predicted by the EOS, see Eq. (1). Thus, all required quantities are obtained using predictive combination rules and mixing rules and the EOS mixture model. No adjustable mixture parameters are introduced. However, for modeling the diffusion coefficients of a given binary system based on a given EOS model, adjustable parameters have to be determined for the four limiting cases, i.e. the two pure components and the two infinite dilution pseudo-pure components, cf. $\alpha$ parameters in Eq. (10). For predicting the diffusion coefficients in the mixture, no adjustable parameters are required. The non-ideality of the mixture is primarily taken into account by the underlying EOS via the predicted configurational mixture entropy. The combination of the EOS and entropy scaling enables the consistent calculation of the diffusion coefficients ($D_i, D_j, Đ_{ij}, D_{ij}$), homogeneous bulk properties ($pvT, s_{\text{conf}}, \Gamma_{ij}$, etc.), and phase equilibria (e.g., vapor-liquid and liquid-liquid equilibria).

## Data availability
The data that support the findings of this study are available from the corresponding author upon request. The molecular simulation data generated in this study are provided as Supplementary Data. Source data are provided in this paper.

## Code availability
The code used in this study is available from the corresponding author upon request. An implementation of the entropy scaling framework is provided as Supplementary Software.

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

## Acknowledgements

We gratefully acknowledge funding from the European Union's Horizon Europe research and innovation program under grant agreement no. 101137725 (BatCAT) and from the German Research Foundation (DFG) under grant 548115878. The simulations were carried out on the HPC machine ELWE at the RHRZ under the grant RPTU-MTD and on the HPC machine MOGON at the NHR SW under the grant TU-MSG (supported by the Federal Ministry of Education and Research and the state governments).

## Author contributions

S. Schmitt: data curation, formal analysis, methodology, software (lead), visualization, writing—original draft. H. H.: funding acquisition, writing—review & editing (support). S. Stephan: conceptualization, methodology, supervision, funding acquisition, software (support), writing—review & editing (lead).

## Funding

## Competing interests

The authors declare no competing interests.
