## [Transparent Peer Review file · Nature Communications]

Entropy Scaling for Diffusion Coefficients in Mixtures

Corresponding Author: Professor Simon Stephan

Version 0:

Reviewer comments:

Reviewer #1

(Remarks to the Author)

The paper 'Entropy Scaling for Diffusion Coefficients in Mixtures' seeks to develop a framework for predicting the diffusion coefficients of mixtures.

I regularly approach this problem from a practical point of view, acquiring and interpreting diffusion coefficients. This general problem is, perhaps surprisingly, neither well covered nor explained. Therefore, a successful, usable, model would be significant and likely to be widely used.

The paper is clear, methodically setting out the different steps and illustrating the results accordingly. The methodology appears sound throughout. There is a great deal of supporting information and data, so the work should be easily reproduced.

My main issue is that the model is tested only twice in the main paper with systems (n-hexane-n-dodecane and n-hexane-toluene) likely to be ideal and do not exhibit much in the way of chemistry. While I am sympathetic to the statement on page 9 that 'practically all real mixtures ... only have little experimental data available', I would suggest the model needs testing against a wider range of mixtures. There's a seam of data compiled in Chemical Engineering literature that may be able to help (suggested starting points of Safi et al in J Chem Eng Data in both 2007 (doi:10.1021/je6005604) and 2008 (doi:10.2021/je700539w), which both have tables and tables of data).

Following on from this, there's a small number of empirical equations that seek to predict diffusion coefficients as a function of various chemical parameters, as well as those that seek to handle mixed solutions and mixed solvents. How well this new model agrees with these antecessors is a question left unanswered by the manuscript and I think should be addressed in some way, even if it is followed up in more detail at a later date by the authors.

Reviewer #2

(Remarks to the Author)

This manuscript reports theoretical and computational work aimed at applying the framework of entropy scaling (ES) to predict the diffusion coefficient of mixtures.

The authors argue that this work is original in that "modeling mixture diffusion coefficients by entropy scaling is an unresolved task". I do not believe this statement to be entirely correct. In Ref.10 [Bell, I.H., Dyre, J.C. & Ingebrigtsen, T.S. Excess-entropy scaling in supercooled binary mixtures. Nat Commun 11, 4300 (2020)], the authors apply an excess-entropy scaling approach which, albeit less general than the one discussed in this paper, can predict diffusion coefficients and other transport properties of supercooled binary mixtures.

More importantly, in Ref. 16 [S. Schmitt, H. Hasse, and S. Stephan, "Entropy scaling framework for transport properties using molecular-based equations of state," Journal of Molecular Liquids 395, 123811 (2024)] the same authors of this manuscript have very recently developed a ES framework "for model fluids as well as for a wide variety of real substances including non-polar, polar, and associating pure fluids and mixtures." The degree of similarity between Ref. 16 and this manuscript is substantial. For instance, Fig.3 in this manuscript (which reports the diffusion coefficients of a Lennard-Jones [LJ] binary

mixtures as a function of composition), is directly related to Fig. 12 in Ref. 16 (which reports the viscosity and the thermal conductivity of a similar LJ mixture).

As the relationship between the work reported in this manuscript and the work of Ref. 16 is only briefly discussed in the present version of the manuscript, I get the impression that the authors are merely applying a modified version of the framework described in Ref.16 (in which transport properties such as thermal conductivity and viscosity were successfully predicted for mixtures containing up to four different components) to expand on the prediction of diffusion coefficients in binary mixtures via ES.

Before discussing the actual robustness of the results presented in this paper, I would urge the authors to elaborate on the significance of Ref. 10 and – crucially – to clarify the relationship between the work reported in Ref. 16 and the work reported in the present manuscript.

At first glance, Ref. 16 and this work apply the same reasoning: the well-established scaling laws for a single-component systems are combined via a set of mixing rules. In fact, if anything, the framework of Ref. 16 appears to be more general than the one described in this paper. Both Ref. 16 and this work utilise equations of state to gather the thermodynamic information necessary to apply the framework: configurational entropy in this work, excess entropy in Ref. 16.

As a further point of feedback, I feel that the current structure of the manuscript is not suitable for a work that revolves around the development and the application of a methodological framework. I appreciate that having the Methods section at the end of the manuscript is common practice for Nature Communications, but in this specific case the Methods are in fact the main content of the work in the first place. As such, I would recommend to re-structure the manuscript to properly discuss the methods section – again, with specific references to Refs 10 and 16.

This is interesting work, but at the moment I fail to see how it differs from an incremental application of a modification of the methodology of Ref. 16 to deal with diffusion coefficients specifically.

Having said that, I would be happy to consider a revised version of this manuscript if the authors were to thoroughly address the points I raised above.

Reviewer #3

(Remarks to the Author)

RE: Entropy scaling for diffusion coefficients in mixtures by Schmitt et al.

This paper addresses the important subject of predicting diffusion coefficients of mixtures and proposes a practical method for doing so, based on entropy scaling. The results appear solid and the paper is generally well written with few (if any) typos. I recommend publication in Nat. Comm. after the authors have had a chance to address the below remarks:

1. Important context - possibly not known to the authors - is missing from the paper. Examples are: a) The study by Samanta et al. [A. Samanta, S. M. Ali, and S. K. Ghosh, Phys. Rev. Lett. 87, 245901 (2001)], which first extended the Dzugutov two-body excess entropy scaling for diffusivity [M. Dzugutov, Nature 381, 137 (1996)] to binary mixtures; b) Pond et al. demonstrated that two-body entropy scaling qualitatively captures the anomalous composition-dependent diffusion trends for soft-particle systems [M. J. Pond, W. P. Krekelberg, V. K. Shen, J. R. Errington, T.M. Truskett, J. Chem. Phys. 131, 161101 (2009)]; c) Krekelberg et al. tested the limits of this approach across several model mixtures and developed a generalized Rosenfeld form of the excess entropy scaling for tracer diffusion [W. P. Krekelberg, M. J. Pond, G. Goel, V. K. Shen, J. R. Errington, T.M. Truskett, Phys. Rev. E 80, 061205 (2009)]; d) Mittal et al. demonstrated Rosenfeld-like excess entropy scalings for tracer diffusion of confined mixtures based on partial molar excess entropies [J. Mittal, V. K. Shen, J. R. Errington, T.M. Truskett, J. Chem. Phys. 127, 154513 (2007)]; e) Carmer et al. used the concept of excess entropy to determine optimal tracer-solvent interactions to enhance colloid tracer diffusivity [J. Carmer, G. Goel, M. J. Pond, J. R. Errington, and T. M. Truskett, Soft Matter 8, 4083-4089 (2012)].

2. I find it a bit frustrating that all details are to be found in the Methods section and Supplement. It must be possible to tell the reader of the main paper, e.g., how many adjustable parameters there are, both in regard to the infinite-dilution diffusion coefficients and in regard to the mixtures.

3. The notation D_{ij} (Eq. (1) etc) with a discrete horizontal line is very unfortunate and would be hard to read if the paper is printed on a poor printer. I urge the authors to use a more robust notation.

4. It should be stated what is kept constant doing the partial derivatives in Eq. (2) (pressure and temperature, presumably).

5. On page 5 the \hat{D} diffusion coefficient should be defined properly. Also it is not clear whether the “binary” mixture is a 50/50 mixture. Generally I recommend the authors to carefully go through the paper to ensure its readability (without having to read the Supplement).

6. In Fig. 3, it is not clear how s_{conf} is defined – is this the \tilde{s} defined later?

7. In Eq. (5) a reference is missing since this is not merely the reduced diffusion coefficient of Rosenfeld and isomorph theory.

8. Below Eq. (7) it is mentioned that W is a “smooth step function” – what is that? Again I find the main paper to be too imprecise.

Reviewer #4

(Remarks to the Author)

This is a really nice application of entropy scaling. It takes advantage of the most up-to-date understanding of how to construct entropy scaling formulations, and demonstrates impressive agreement with experimental (and simulation) results. The work was clearly carried out with great care based on the extremely nice figures, carefully worded manuscript (some small edits notwithstanding), and detailed SI.

I have made some small editorial edits in the attached PDF and have some more general comments here:

- * I am agnostic about whether you call it internal entropy, excess entropy, residual entropy, or configurational entropy. Nevertheless, each of these terms have several possible definitions depending on the technical discipline so you need to define the term explicitly, like so: $s_{\text{conf}} = s(T, \rho) - \text{sig}(T, \rho)$
- * So too the modified entropy scaling terms with the \wedge above need to be defined at first use
- * The modified entropy scaling approach was first proposed by Rosenfeld in 1999 for a narrow application, “rediscovered” by Dyre in his review, and practically used for the first time for “normal” fluids in two concurrent publications (<https://www.pnas.org/doi/abs/10.1073/pnas.1815943116> and <https://pubs.acs.org/doi/full/10.1021/acs.jpcb.9b05808>). It would be appropriate to include that history, the text reads as though you discovered the modified entropy scaling approach since you only refer to your previous work on the topic when describing the model formulation. So too, the approach used to model the individual diffusion terms is based on that in <https://pubs.acs.org/doi/full/10.1021/acs.jpcb.9b05808> but that is not mentioned. What is the form of the W , the smoothed step function? It appears to be the same as in <https://pubs.acs.org/doi/full/10.1021/acs.jpcb.9b05808>? There are many possible smoothed step functions (more generally, “sigmoidal” functions)
- * How does one take the minimum of a function $D_{\text{CE},i}(\infty, \rho)$ that depends on temperature (as in Eq. 7)? Digging into the SI I see it, but you might want to comment in the main manuscript.
- * I am confused about the use of the Kolafa-Nezbeda EOS to obtain Γ_{ij} . How do you do that from an EOS for a pure fluid? For instance, in Figure 2?
- * You might want to be aware that dividing the entropy term by the segment number m is very similar to dividing by the entropy term at the critical point
- * Thank you for providing such a detailed implementation of your model in Julia; you should be commended for doing so. I wasn’t able to find a practical example of calling the function given temperature and density inputs, so I don’t know how I am supposed to use this model. Please provide a runnable example linked with one of the open-source EOS libraries.
- * The MD results in the SI have column headings that are different than the nomenclature used in the paper. Please add a README describing the column headings, or add a header within the data files describing the column headers
- * I would like to see some quantitative comparisons in the main manuscript. They are present in the SI, but you have to go digging to find them.
- * Results for only a few systems are shown in the main manuscript. Based on those results, one might be led to believe that ALL systems would be modeled with the same level of fidelity. I cannot believe that to be true. It might be good to show some of the less-good results too in order to paint a more balanced picture. In a similar vein, your model is much more accurate than the other models you compare it against, and your results make a compelling case for your model becoming the go-to approach, so why not show the other models in the main manuscript too? The figure with all the models overlaid (Fig #7 in the SI) paints the picture quite clearly and could replace the existing figure. You could make your model have thicker lines, or make the other models have transparent curves to differentiate while keeping the same color scheme.

Editorial comments:

- * A pedantic comment is that “lines” must be straight while “curves” need not be straight. So, most of the time where you write “line” you want instead “curve”
- * Since your SI is rather massive (but thank you for providing such a thorough SI!), you might want to consider to cross-reference the section or figure of the SI. In LaTeX you can use the `zref-xr` package to keep references to the SI in sync. You put something like this in the SI file: `\usepackage{zref-xr}` and this in the main document:

```
\usepackage{zref-xr,zref-user}
\zxrsetup{verbose}
\externaldocument*{SI_filename}
```

Ian Bell, NIST

Version 1:

Reviewer comments:

Reviewer #1

(Remarks to the Author)

I have re-read the manuscript, with reviewers comments in mind.

Authors have successfully addressed all points raised in my comments.

On the third comment, I was thinking of equations like Wilke-Chang and Stokes-Einstein-Gierer-Wirtz, but am satisfied with the response.

Thank you.

(Remarks on code availability)

Reviewer #2

(Remarks to the Author)

The authors have thoroughly addressed all my comments, making the relationship between this work and past efforts in similar directions much clearer.

The few technical concerns I initially had (most notably the need to apply the framework to some more realistic systems) were already raised by the other reviewers, and I believe the authors have done an excellent job addressing them.

The revised manuscript is robust and reads very well. While the journal's format, particularly placing the Methods section at the end, does not best suit the presentation of this work, I understand this cannot be changed.

I am happy to recommend this version of the manuscript for publication in Nature Communications.

(Remarks on code availability)

Reviewer #3

(Remarks to the Author)

I have no objections to the paper being published by NC in its revised version. The authors seem to have answered the many points raised by all reviewers.

(Remarks on code availability)

Reviewer #4

(Remarks to the Author)

Thank you for your edits which have improved the quality of the manuscript. I take issue with the provided code. The EOS implementation is not open-source (there are plenty of good open-source implementations that could be used), and is furthermore not available on non-windows operating systems based on what was provided in the SI. Please provide the code used for the EOS implementation. Since you wrote the model in Julia, it would be most straightforward to use Clapeyron.jl.

(Remarks on code availability)

Not reproducible. The EOS implementation is provided as compiled files, which is not acceptable, and furthermore, the provided installer only works on windows operating systems, so not linux or macos. As a mac user, I cannot use their code.

Response to reviewer's comments (NCOMMS-24-58186)

Times: reviews (full text)

Courier: response (page and line numbers given in the responses refer to the "Revised Manuscript"). The changes in the "Revised Manuscript" are highlighted by red text color.

Reviewer #1

The paper 'Entropy Scaling for Diffusion Coefficients in Mixtures' seeks to develop a framework for predicting the diffusion coefficients of mixtures.

I regularly approach this problem from a practical point of view, acquiring and interpreting diffusion coefficients. This general problem is, perhaps surprisingly, neither well covered nor explained. Therefore, a successful, usable, model would be significant and likely to be widely used.

The paper is clear, methodically setting out the different steps and illustrating the results accordingly. The methodology appears sound throughout. There is a great deal of supporting information and data, so the work should be easily reproduced.

We thank the reviewer for the kind appraisal and share the reviewer's view on the problem.

My main issue is that the model is tested only twice in the main paper with systems (n-hexane-n-dodecane and n-hexane-toluene) likely to be ideal and do not exhibit much in the way of chemistry. While I am sympathetic to the statement on page 9 that 'practically all real mixtures ... only have little experimental data available', I would suggest the model needs testing against a wider range of mixtures. There's a seam of data compiled in Chemical Engineering literature that may be able to help (suggested starting points of Safi et al in J Chem Eng Data in both 2007 (doi:10.1021/je6005604) and 2008 (doi:10.2021/je700539w), which both have tables and tables of data).

Done. We have more than doubled the amount of test systems. Additional tests were included for a wider range of mixtures. The new results support the general applicability of the framework. However, it is out of the scope of this paper to extend out new approach to ternary and multi-component mixtures. This is not trivial, but we are planning on addressing this in future work. We have added a comment on this in the Conclusion.

Following on from this, there's a small number of empirical equations that seek to predict diffusion coefficients as a function of various chemical parameters, as well as those that seek to handle mixed solutions and mixed solvents. How well this new model agrees with

these antecessors is a question left unanswered by the manuscript and I think should be addressed in some way, even if it is followed up in more detail at a later date by the authors.

We thank the reviewer for this comment. We have added a brief discussion on the comparison of our new model with two empirical models, namely the Vignes model and the generalized Darken model (see page 9 of the revised manuscript). The corresponding data is presented in the SI. Our entropy scaling model significantly outperforms the two empirical models.

Reviewer #2

This manuscript reports theoretical and computational work aimed at applying the framework of entropy scaling (ES) to predict the diffusion coefficient of mixtures.

The authors argue that this work is original in that “modeling mixture diffusion coefficients by entropy scaling is an unresolved task”. I do not believe this statement to be entirely correct. In Ref.10 [Bell, I.H., Dyre, J.C. & Ingebrigtsen, T.S. Excess-entropy scaling in supercooled binary mixtures. Nat Commun 11, 4300 (2020)], the authors apply an excess-entropy scaling approach which, albeit less general than the one discussed in this paper, can predict diffusion coefficients and other transport properties of supercooled binary mixtures.

We thank the reviewer for this comment. As correctly stated, the mentioned Ref. 10 from Bell et al. also deals with entropy scaling of diffusion coefficients and demonstrates the general scaling behavior diffusion coefficient data in mixtures, i.e. the fact that mixture diffusion coefficients may in some special cases yield a monovariate function (as also done in other references). However, these references do not provide a general approach to the modeling (!) of the diffusion coefficients and, additionally, they do not address mutual diffusion coefficients. We have modified the wording in the manuscript to clarify what has been done in the literature as well as the highlighting the fundamentally new aspects of our approach:

“Furthermore, a quasi-universal scaling law for mixture diffusion coefficients has been proposed by Bell and Dyre¹⁰ – however, being limited to self-diffusion. Truskett and co-workers²⁹⁻³² have studied the monovariate scaling behavior based on computer experiment data. Similarly, Fertig et al.³³ have demonstrated that elements of monovariate scaling are also present in model mixture diffusion coefficient

data. However, in none of these studies a generally applicable modeling framework for consistently describing the different diffusion coefficients in mixtures has been developed, so that this is still an unresolved issue.” (page 4 of the revised manuscript)

More importantly, in Ref. 16 [S. Schmitt, H. Hasse, and S. Stephan, “Entropy scaling framework for transport properties using molecular-based equations of state,” *Journal of Molecular Liquids* 395, 123811 (2024)] the same authors of this manuscript have very recently developed a ES framework “for model fluids as well as for a wide variety of real substances including non-polar, polar, and associating pure fluids and mixtures.” The degree of similarity between Ref. 16 and this manuscript is substantial. For instance, Fig.3 in this manuscript (which reports the diffusion coefficients of a Lennard-Jones [LJ] binary mixtures as a function of composition), is directly related to Fig. 12 in Ref. 16 (which reports the viscosity and the thermal conductivity of a similar LJ mixture).

As the relationship between the work reported in this manuscript and the work of Ref. 16 is only briefly discussed in the present version of the manuscript, I get the impression that the authors are merely applying a modified version of the framework described in Ref.16 (in which transport properties such as thermal conductivity and viscosity were successfully predicted for mixtures containing up to four different components) to expand on the prediction of diffusion coefficients in binary mixtures via ES.

This comment surprised us, but we take it as a sign that, indeed, the relationship between this work and our previous one (which is very clearly delineated) does indeed need to be discussed more explicitly. In our previous work (formerly Ref. 16, now Ref. 17), we have proposed a way of entropy scaling that can - in particular - be favorably combined with molecular-based equations of state and have applied it to different pure component and mixture properties, but we have NOT applied it to modeling diffusion coefficients in mixtures - as this cannot be done without introducing additional new concepts - which is exactly what we have done in the present work.

The first main new idea of the present work is to apply entropy scaling to the two limiting diffusion coefficients. Once it is shown that this works (as we have done in the current work), the idea can be combined with the (previously existing) knowledge on the pure component self-diffusion coefficients and, then, a way has to be found to extend the concepts to mixtures with finite concentrations of the components. This is where the second major new idea of our current work comes in: the concept of applying mixing and combination rules for bringing the pure component and pseudo-pure component models together for

predicting all types of diffusion coefficients in the mixtures in a consistent framework.

None of this has been addressed at all in our previous work nor in any other work in the literature.

However, we have used the ideas from our previous work to find correlations for the limiting infinite dilution diffusion coefficients, which is clearly stated in the manuscript. Moreover, the modeling approach proposed in this work is generic and can also be used with other entropy scaling frameworks than the one proposed in Ref. 16, e.g. the one proposed by Gross and co-workers, as well as be coupled with other EOS, which are not of the molecular-based EOS type. We have clarified this in the manuscript as:

"The modeling methodology is described in the following. There are different entropy scaling approaches available in the literature that aim at establishing a monovariate behavior of transport coefficients. Here, we use the scaling described in Ref. 17. The pure component scaling described in Ref. 17 (which follows Refs. 13,16) is adapted here to model the pseudo-pure component diffusion coefficients at infinite dilution." (page 16 of the revised manuscript)

and

"In this work, we use molecular-based EOS models. Yet, also other EOS types such as multiparameter³⁴ or cubic EOS³⁵ could be used." (page 5 of the revised manuscript)

Before discussing the actual robustness of the results presented in this paper, I would urge the authors to elaborate on the significance of Ref. 10 and – crucially – to clarify the relationship between the work reported in Ref. 16 and the work reported in the present manuscript.

This summarizes issues that were raised in previous comments of the reviewer. The relationship of our work to Ref. 16 is discussed in the answer above, the relationship to Ref. 10 is discussed in our answer to the first comment of the reviewer. The corresponding statements in the paper were revised for clarity.

At first glance, Ref. 16 and this work apply the same reasoning: the well-established scaling laws for a single-component systems are combined via a set of mixing rules. In

fact, if anything, the framework of Ref. 16 appears to be more general than the one described in this paper. Both Ref. 16 and this work utilise equations of state to gather the thermodynamic information necessary to apply the framework: configurational entropy in this work, excess entropy in Ref. 16.

The reviewer again raises the question of similarities of Ref. 16 (Ref. 17 in the revised manuscript) and the manuscript under review. The two differ fundamentally, see our response above. We build on our previous work, but could have used also other entropy scaling and EOS approaches for our new diffusion modeling model. The work under review enables for the very first time the thermodynamic consistent modeling of all diffusion coefficients in a binary mixture. The fact that four different diffusion coefficients (two self-diffusion and two mutual diffusion coefficients) are described by our new approach highlights the fundamental differences to established mixture entropy scaling for the thermal conductivity and viscosity (which were applied in Ref. 16). Nothing of that kind is addressed in Ref. 16.

We consistently use the term "configurational entropy" - both here and in previous works of our group such as Ref. 16. Sometimes, the terms "residual entropy" and "excess entropy" are used in the literature. We have added a comment in the revised manuscript (page 4) that other terms are at times used in the literature.

As a further point of feedback, I feel that the current structure of the manuscript is not suitable for a work that revolves around the development and the application of a methodological framework. I appreciate that having the Methods section at the end of the manuscript is common practice for Nature Communications, but in this specific case the Methods are in fact the main content of the work in the first place. As such, I would recommend to re-structure the manuscript to properly discuss the methods section - again, with specific references to Refs 10 and 16.

In general, we agree with the reviewers point on the manuscript structure. The complete scaling procedure is now described in detail in the main part. However, after consulting the editor, we have decided to follow the journal guidelines and keep the structure as is. We have clarified the relation of the manuscript under review and Refs. 10 and 16 (and other work in the literature). See our responses above and pages 5 and 16 in the revised manuscript.

"The monovariate scaling of the infinite dilution diffusion coefficients uncovered in this work provides the basis for the modeling of the different mixture diffusion coefficients shown in the second part of this chapter. The applicability of the new modeling approach is demonstrated in the second part of this chapter, where model predictions for the self-diffusion as well as mutual diffusion coefficients are compared to reference data for model and real substance systems." (page 5 of the revised manuscript)

This is interesting work, but at the moment I fail to see how it differs from an incremental application of a modification of the methodology of Ref. 16 to deal with diffusion coefficients specifically.

Having said that, I would be happy to consider a revised version of this manuscript if the authors were to thoroughly address the points I raised above.

We thank the reviewer for this open feedback and the willingness to review also the revision. We appreciate his comments and have taken advantage of them to improve the manuscript. The reviewer seems to be preliminarily concerned with putting our work in context with the two references 10 and 16 (from the original submission). We have clarified this at several points in the manuscript.

Reviewer #3

This paper addresses the important subject of predicting diffusion coefficients of mixtures and proposes a practical method for doing so, based on entropy scaling. The results appear solid and the paper is generally well written with few (if any) typos. I recommend publication in Nat. Comm. after the authors have had a chance to address the below remarks:

We thank the reviewer for the positive evaluation of our work.

1. Important context - possibly not known to the authors - is missing from the paper. Examples are:
 - a. The study by Samanta et al. [A. Samanta, S. M. Ali, and S. K. Ghosh, Phys. Rev. Lett. 87, 245901 (2001)], which first extended the Dzugutov two-body excess entropy scaling for diffusivity [M. Dzugutov, Nature 381, 137 (1996)] to binary mixtures;
 - b. Pond et al. demonstrated that two-body entropy scaling qualitatively captures the anomalous composition-dependent diffusion trends for soft-particle systems [M. J. Pond, W. P. Krekelberg, V. K. Shen, J. R. Errington, T.M. Truskett, J. Chem. Phys. 131, 161101 (2009)];

- c. Krekelberg et al. tested the limits of this approach across several model mixtures and developed a generalized Rosenfeld form of the excess entropy scaling for tracer diffusion [W. P. Krekelberg, M. J. Pond, G. Goel, V. K. Shen, J. R. Errington, T.M. Truskett, Phys. Rev. E 80, 061205 (2009)];
- d. Mittal et al. demonstrated Rosenfeld-like excess entropy scalings for tracer diffusion of confined mixtures based on partial molar excess entropies [J. Mittal, V. K. Shen, J. R. Errington, T.M. Truskett, J. Chem. Phys. 127, 154513 (2007)];
- e. Carmer et al. used the concept of excess entropy to determine optimal tracer-solvent interactions to enhance colloid tracer diffusivity [J. Carmer, G. Goel, M. J. Pond, J. R. Errington, and T. M. Truskett, Soft Matter 8, 4083-4089 (2012)].

We thank the reviewer for bringing these papers on self-diffusion in mixtures to our attention. We have added a short corresponding discussion context:

"There are few molecular simulation studies of self-diffusion coefficients in mixtures available in the literature^{10,27-33} in which also the entropy scaling behavior of the data was considered. They evaluate either simple model systems or special cases like metallic fluids. They show that self-diffusion coefficients of such fluids can follow a universal monovariate behavior, but the authors do not provide models for describing the self-diffusion coefficients." (page 4 of the revise manuscript)

2. I find it a bit frustrating that all details are to be found in the Methods section and Supplement. It must be possible to tell the reader of the main paper, e.g., how many adjustable parameters there are, both in regard to the infinite-dilution diffusion coefficients and in regard to the mixtures.

We agree and have modified the manuscript such that the complete modeling procedure is described in the main part. We have also added a paragraph summarizing where adjustable parameters occur in the model.

3. The notation D_{ij} (Eq. (1) etc) with a discrete horizontal line is very unfortunate and would be hard to read if the paper is printed on a poor printer. I urge the authors to use a more robust notation.

We believe the reviewer is referring to the notation of the Maxwell-Stefan diffusion coefficient. As the notation used in this work is the common notation of the Maxwell-Stefan diffusion coefficient, we would like to keep it. However, we have made sure that the

variable D_{ij} with the horizontal line is accompanied with the name 'Maxwell-Stefan diffusion coefficient' to avoid confusion.

4. It should be stated what is kept constant doing the partial derivatives in Eq. (2) (pressure and temperature, presumably).

Done. Eq. (2) was modified accordingly in the revised manuscript.

5. On page 5 the \hat{D} diffusion coefficient should be defined properly. Also it is not clear whether the "binary" mixture is a 50/50 mixture. Generally I recommend the authors to carefully go through the paper to ensure its readability (without having to read the Supplement).

We understand the points raised by the reviewer, which mainly refer to the clarity of terms in the main manuscript and have accordingly revised the manuscript at multiple points. In particular, we have added an explaining section, in which the different elements of our notation are explained, cf. page 5 of the revised manuscript.

6. In Fig. 3, it is not clear how s_{conf} is defined – is this the s_{tilde} defined later?

We have added the definitions of s_{conf} and s_{tilde} in the Methods section. We thank the reviewer for pointing this out to us.

7. In Eq. (5) a reference is missing since this is not merely the reduced diffusion coefficient of Rosenfeld and isomorph theory.

Done. We have revised this text and added the references that introduced the modified scaling.

8. Below Eq. (7) it is mentioned that W is a "smooth step function" – what is that? Again I find the main paper to be too imprecise.

In the revised manuscript, the definition of W is part of the main body of the paper. We have changed the wording to 'smooth transition function'.

Reviewer #4

This is a really nice application of entropy scaling. It takes advantage of the most up-to-date understanding of how to construct entropy scaling formulations, and demonstrates impressive agreement with experimental (and simulation) results. The work was clearly

carried out with great care based on the extremely nice figures, carefully worded manuscript (some small edits notwithstanding), and detailed SI.

I have made some small editorial edits in the attached PDF and have some more general comments here:

We thank the reviewer for the kind appraisal and the thorough reading.

- I am agnostic about whether you call it internal entropy, excess entropy, residual entropy, or configurational entropy. Nevertheless, each of these terms have several possible definitions depending on the technical discipline so you need to define the term explicitly, like so: $s_{\text{conf}} = s(T, \rho) - \text{sig}(T, \rho)$

Done. We have added the definition of s_{conf} to the manuscript.

- So too the modified entropy scaling terms with the \wedge above need to be defined at first use

Done. We have made changes such that variables are properly introduced to the reader upon their first appearance - and give reference to the Methods section, where the mathematical definitions are given. Moreover, in the revised manuscript, we have moved all essential parts of the equations to the main body (see comment from Reviewer 2).

- The modified entropy scaling approach was first proposed by Rosenfeld in 1999 for a narrow application, “rediscovered” by Dyre in his review, and practically used for the first time for “normal” fluids in two concurrent publications (<https://www.pnas.org/doi/abs/10.1073/pnas.1815943116> and <https://pubs.acs.org/doi/full/10.1021/acs.jpcc.9b05808>). It would be appropriate to include that history, the text reads as though you discovered the modified entropy scaling approach since you only refer to your previous work on the topic when describing the model formulation. So too, the approach used to model the individual diffusion terms is based on that in <https://pubs.acs.org/doi/full/10.1021/acs.jpcc.9b05808> but that is not mentioned. What is the form of the W , the smoothed step function? It appears to be the same as in <https://pubs.acs.org/doi/full/10.1021/acs.jpcc.9b05808>? There are many possible smoothed step functions (more generally, “sigmoidal” functions)

We have added a brief discussion on the history of entropy scaling in the Introduction. The references pointed out by the reviewer are now cited in our manuscript accordingly. The references for the

modified entropy scaling approach were included and their contribution to the scaling applied in this work highlighted (see also above).

"After being re-discovered by Dyre in a seminal review¹¹, entropy scaling has become a popular approach in recent years. For modeling transport properties, the entropy scaling principle has been cast into many executable models using several different modified scaling approaches^{9,15-19}." (page 4 of the revised manuscript)

We have also added the equation for the function W to the main part of the revised manuscript.

- How does one take the minimum of a function $D_{(CE,i)}^{(\infty,0)}$ that depends on temperature (as in Eq. 7)? Digging into the SI I see it, but you might want to comment in the main manuscript.

The corresponding parts of the SI have now been included in the main part of the paper, such that the complete procedure is precisely defined to the reader.

- I am confused about the use of the Kolafa-Nezbeda EOS to obtain Γ_{ij} . How do you do that from an EOS for a pure fluid? For instance, in Figure 2?

In a previous work of our group, we have applied the KN-EOS for the modeling of binary mixtures. This was done using van der Waals one fluid theory and adapting the modified Lorentz-Berthelot combination rules. We have added a brief comment on this, cf. page 8 of the revised manuscript.

- You might want to be aware that dividing the entropy term by the segment number m is very similar to dividing by the entropy term at the critical point

We thank the reviewer for this interesting comment, and agree. Both approaches, scaling the entropy by the segment number of EOS and scaling by the critical entropy serve the same purpose to make the scaling behavior of different substances similar. We have clarified this when introducing s_{tilde} , cf. page 16.

- Thank you for providing such a detailed implementation of your model in Julia; you should be commended for doing so. I wasn't able to find a practical example of calling the function given temperature and density inputs, so I don't know how

I am supposed to use this model. Please provide a runnable example linked with one of the open-source EOS libraries.

We thank the reviewer for this very helpful suggestion. A runnable example is now provided in the electronic Supporting Information.

- The MD results in the SI have column headings that are different than the nomenclature used in the paper. Please add a README describing the column headings, or add a header within the data files describing the column headers

Done. The headers were revised to match the nomenclature of the paper.

- I would like to see some quantitative comparisons in the main manuscript. They are present in the SI, but you have to go digging to find them.

Done. We have added quantitative comparisons (AAD values) between the entropy scaling model and reference data for all systems in the main part.

- Results for only a few systems are shown in the main manuscript. Based on those results, one might be led to believe that ALL systems would be modeled with the same level of fidelity. I cannot believe that to be true. It might be good to show some of the less-good results too in order to paint a more balanced picture. In a similar vein, your model is much more accurate than the other models you compare it against, and your results make a compelling case for your model becoming the go-to approach, so why not show the other models in the main manuscript too? The figure with all the models overlaid (Fig #7 in the SI) paints the picture quite clearly and could replace the existing figure. You could make your model have thicker lines, or make the other models have transparent curves to differentiate while keeping the same color scheme.

Done. We have added the results for more systems to the manuscript and included a critical discussion. We have practically doubled the number of systems where we apply the new model. Also, we have added results for a system where the model does not perform as good, namely a binary mixture with an associating component. Moreover, we have revised the respective figure (now Fig. S6) in the SI according to the suggestions.

Editorial comments:

- A pedantic comment is that “lines” must be straight while “curves” need not be straight. So, most of the time where you write “line” you want instead “curve”

We understand the reviewer's comment. However, we would like to keep the term 'line' as it is the usually used in this context. This is supported by the definition in the Oxford dictionary.

Nevertheless, we appreciate the reviewer for pointing this out to us since we fully agree that a rigorous language is essential in science.

- Since your SI is rather massive (but thank you for providing such a thorough SI!), you might want to consider to cross-reference the section or figure of the SI. In LaTeX you can use the zref-xr package to keep references to the SI in sync. You put something like this in the SI file: `\usepackage{zref-xr}` and this in the main document:

```
\usepackage{zref-xr,zref-user}
\zxrsetup{verbose}
\zexternaldocument*{SI_filename}
```

Done. We thank the reviewer for this comment. We have added cross-references to the SI in the main part.

Response to reviewer's comments (NCOMMS-24-58186)

Times: reviews (full text)

Courier: response (page and line numbers given in the responses refer to the "Revised Manuscript"). The changes in the "Revised Manuscript" are highlighted by red text color.

Reviewer #1

I have re-read the manuscript, with reviewers comments in mind.

Authors have successfully addressed all points raised in my comments.

On the third comment, I was thinking of equations like Wilke-Chang and Stokes-Einstein-Gierer-Wirtz, but am satisfied with the response.

Thank you.

We thank the reviewer for the positive evaluation.

An in-depth comparison of our new approach with models from the literature like Wilke-Chang and Stokes-Einstein-Gierer-Wirtz is planned for a future separate work.

Reviewer #2

The authors have thoroughly addressed all my comments, making the relationship between this work and past efforts in similar directions much clearer.

The few technical concerns I initially had (most notably the need to apply the framework to some more realistic systems) were already raised by the other reviewers, and I believe the authors have done an excellent job addressing them.

The revised manuscript is robust and reads very well. While the journal's format, particularly placing the Methods section at the end, does not best suit the presentation of this work, I understand this cannot be changed.

I am happy to recommend this version of the manuscript for publication in Nature Communications.

We thank the reviewer for the positive evaluation.

Reviewer #3

I have no objections to the paper being published by NC in the revised version. The authors seem to have answered the many points raised by all reviewers.

We thank the reviewer for the positive evaluation.

Reviewer #4

Thank you for your edits which have improved the quality of the manuscript. I take issue with the provided code. The EOS implementation is not open-source (there are plenty of good open-source implementations that could be used), and is furthermore not available on non-windows operating systems based on what was provided in the SI. Please provide the code used for the EOS implementation. Since you wrote the model in Julia, it would be most straightforward to use Clapeyron.jl.

Remarks on code availability

Not reproducible. The EOS implementation is provided as compiled files, which is not acceptable, and furthermore, the provided installer only works on windows operating systems, so not linux or macos. As a mac user, I cannot use their code.

We thank the reviewer for the positive feedback on our revision.

We have revised the appended code and included the Julia implementations of the applied EOS such that the employed code is now available. This enables the execution of the code platform-independent and does not require the installation of a binary file.